

# Meteorological modeling sensitivity to parameterizations and

# satellite-derived surface datasets during the 2017 Lake Michigan

# Ozone Study

Jason A. Otkin[1,2], Lee M. Cronce[1,2], Jonathan L. Case[3], R. Bradley Pierce[1], Monica Harkey[4], Allen
Lenzen[1], David S. Henderson[1], Zac Adelman[5], Tsengel Nergui[5], Christopher R. Hain[6]
[1]Space Science and Engineering Center, University of Wisconsin-Madison, Madison, 53706, USA
[2]Cooperative Institute for Meteorological Satellite Studies, University of Wisconsin, Madison, Madison, 53706, USA
[3]ENSCO, Inc., NASA Short-term Prediction Research and Transition Center, Huntsville, 35805, USA
[4]Center for Sustainability and the Global Environment, University of Wisconsin-Madison, Madison, 53706, USA
[5]Lake Michigan Air Directors Consortium, Hillside, 60162, USA
[6]Earth Science Office, NASA Marshall Space Flight Center, Huntsville, 35808, USA

*Correspondence to*: Jason A. Otkin (jasono@ssec.wisc.edu)

**Abstract.** High-resolution simulations were performed to assess the impact of different parameterization schemes, surface
initialization datasets, and analysis nudging on lower-tropospheric conditions near Lake Michigan. Simulations were run
where climatological or coarse-resolution surface initialization datasets were replaced by high-resolution, real-time datasets
depicting lake surface temperatures (SST), green vegetation fraction (GVF), and soil moisture and temperature (SOIL).
Comparison of a baseline simulation employing a configuration similar to that used at the Environmental Protection Agency
("EPA") to another simulation employing an alternative set of parameterization schemes (referred to as "YNT") showed that
the EPA configuration produced more accurate analyses on the outermost 12-km resolution domain, but that the YNT
configuration was superior for higher-resolution nests. The diurnal evolution of the surface energy fluxes was similar in both
simulations on the 12-km grid but differed greatly on the 1.3-km grid where the EPA simulation had much smaller sensible
heat flux during the daytime and physically unrealistic ground heat flux. Switching to the YNT configuration led to substantial
decreases in root mean square error for 2-m temperature and 2-m water vapor mixing ratio on the 1.3-km grid. Additional
improvements occurred when the high-resolution satellite-derived surface datasets were incorporated into the modeling
platform, with the SOIL dataset having the largest positive impact on temperature and water vapor. The GVF and SST datasets
also produced more accurate temperature and water vapor analyses, but degradations in wind speed, especially when using the
GVF dataset. The most accurate simulations were obtained when using the high-resolution SST and SOIL datasets and analysis
nudging above 2 km AGL.

## 1 Introduction

Locations along the Lake Michigan shoreline in the United States have a long history of recording surface ozone concentrations
that exceed levels set by the National Ambient Air Quality Standards (NAAQS), especially during the warm season (Stanier
et al. 2021). Since the first ozone NAAQS was released in 1979, most lakeshore counties in the states bordering Lake Michigan
(Wisconsin, Illinois, Indiana, and Michigan) have been designated as being in nonattainment for surface ozone in one or more



of the subsequent NAAQS revisions. These states are required by the Clean Air Act to develop State Implementation Plans
(SIPs) to demonstrate strategies to bring affected areas into attainment and to mitigate the impacts of high ozone
concentrations. Large decreases in local emissions of ozone precursors such as nitrogen oxides and volatile organic compounds
have steadily reduced one- and eight-hour maximum ozone concentrations across the region in recent decades (Adelman 2020).
However, the implementation of stricter ozone NAAQS means that additional air quality modeling assessments are necessary
to help states demonstrate that they can reach attainment by the required statutory deadlines.
Urban and rural areas near Lake Michigan are susceptible to high ozone events due to the complex interaction between synoptic
and mesoscale circulation patterns with large sources of industrial, transportation, and urban emissions along the southern end
of the lake. High ozone days are most common when synoptic-scale weather patterns characterized by weak southerly winds
transport ozone and its precursors northward from their primary source regions over the Chicago and Milwaukee metropolitan
areas and then interact with the mesoscale lake and land breeze circulations (Lyons and Olsson 1973; Ragland and Samson
1977; Lennartson and Schwartz 2002). At night, the land breeze carries ozone precursors from land-based emissions sources
over the lake where they become confined within a shallow nocturnal boundary layer and are then converted into ozone after
sunrise via photochemical processes (Dye et al. 1995). As the land surface warms during the day, a reversal of the mesoscale
circulation leads to the formation of the lake breeze during the morning that transports the high ozone airmass back onshore,
with elevated ozone concentrations occurring across inland areas during midday and afternoon. On high ozone days, the lowest
ozone concentrations are often found in areas with high nitrogen oxide emissions, such as Chicago and northwestern Indiana,
with the highest ozone levels located downwind in rural and suburban areas to the north of these urban and industrial locations
(Foley et al. 2011; Cleary et al. 2015).
When synoptic-scale conditions are favorable for lake and land breeze formation, the horizontal temperature gradient between
adjacent land and water areas influences the strength of the circulation pattern and the distance that the lake breeze penetrates
inland during the daytime. Changes in the location of the lake breeze can have a profound impact on near-surface meteorology,
the depth and vertical structure of the planetary boundary layer (PBL), and ozone concentrations along the Lake Michigan
shoreline (Dye et al. 1995). Among other things, an accurate depiction of near-surface features in numerical weather prediction
models requires an accurate specification of lower boundary conditions at the land and water surface. For example, an accurate
representation of land surface conditions (such as soil moisture, soil temperature, and green vegetation fraction) are necessary
to correctly partition the surface net radiation into sensible, latent, and ground heat fluxes. This partitioning in turn impacts
the growth and depth of the PBL and lower-tropospheric temperature, moisture, and wind profiles (Berg et al. 2014; Dirmeyer
and Halder 2016; Schwingshakl et al. 2017; Welty and Zeng 2018). Soil moisture and vegetation fraction (or leaf area index)
are especially important variables through their influence on land-atmosphere coupling processes that link the surface
hydrologic and atmospheric components of the earth system (Santanello et al. 2018, 2019). Indeed, Huang et al. (2017) showed
that use of improved soil moisture and green vegetation fraction estimates in high-resolution simulations reduced biases in
near-surface air temperatures and PBL heights over the Missouri Ozarks and had a large impact on biogenic isoprene
emissions.
Given the important role that boundary layer meteorology and the land-lake breeze circulation have on ozone production and
transport in the Lake Michigan region, it is critical to explore the ability of different parameterization schemes and surface
initialization datasets to improve the accuracy of near-surface meteorological and air quality simulations. In this two-part
study, we develop and assess the accuracy of a satellite-constrained modeling platform for the Midwest United States that
supports the needs of the Lake Michigan Air Directors Consortium (LADCO) as they conduct detailed air quality modeling
assessments for its member states. The modeling platform uses high-resolution analyses of soil moisture, green vegetation
fraction, and lake surface temperatures derived from satellite observations and an offline land surface model (LSM) to
constrain the evolution of the lower boundary conditions during multi-week model simulations. In part I, we use results from
a large set of Weather Research and Forecasting (WRF) model simulations to assess the impact of the high-resolution surface
datasets, different parameterization schemes, and analysis nudging on near-surface meteorological conditions and energy
fluxes. We will show that a baseline model configuration employing surface datasets and parameterization schemes similar to
those used by the United States Environmental Protection Agency (EPA) produces better results for model simulations
performed at 12-km horizontal grid spacing, but that more accurate results are obtained at higher resolutions when the satellite-





derived initialization datasets and alternative parameterization schemes are used. In part II of this study, we use meteorological
analyses obtained from the baseline EPA and optimized WRF model configurations as input to Community Multiscale Air
Quality (CMAQ) model simulations to assess the impact of these model changes on ozone forecasts in the Lake Michigan
region. The remainder of this paper is organized as follows. Section 2 contains a description of the model configurations and
surface initialization datasets. Results are presented in Section 3, with a discussion and conclusions provided in Section 4.

**2. Methods**

**2.1 WRF model configurations**

Version 3.8.1 of the WRF model (Powers et al. 2017) was used to perform simulations containing three one-way nested
domains covering the contiguous United States, Midwest United States, and Lake Michigan regions with 12, 4, and 1.3 km
horizontal resolutions, respectively (Fig. 1). Each simulation contained 40 terrain-following vertical layers, with the model top
set to 100 hPa. The 0.25-degree resolution GFS Final reanalyses available at 6-h intervals served as initial and lateral boundary
conditions (ICs/BCs) for the WRF simulations. All simulations were run from 12 May 2017 – 22 June 2017, with our analysis
focusing on the 22 May – 22 June 2017 time period corresponding to the Lake Michigan Ozone Study field project (Stainer et
al. 2021).

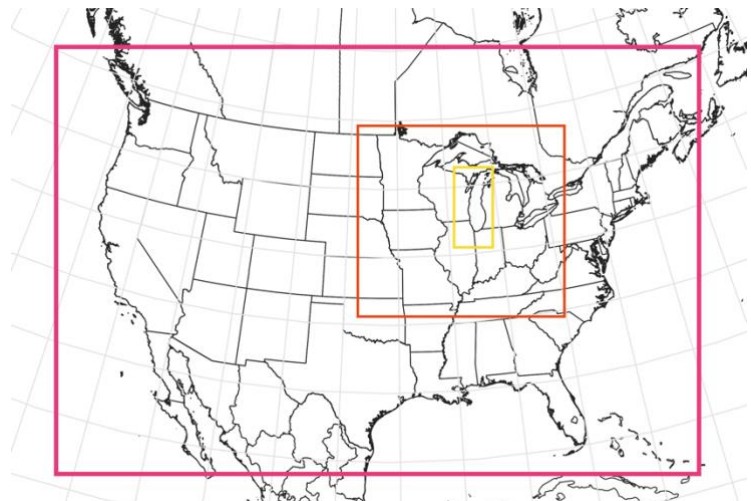

**Figure 1. Map showing the geographic regions covered by the 12-km (red box), 4-km (orange box), and 1.3-km (yellow box)**
**resolution domains used during the WRF model experiments.**
Eight model simulations were performed to assess the impact of different physics options and surface initialization datasets
on the model accuracy in the lower troposphere (Table 1). The first simulation employed a configuration similar to that used
in air quality modeling at the EPA and is hereafter referred to as the "EPA" baseline configuration. This simulation employed
the Morrison microphysics (Morrison et al. 2005), RRTMG longwave and shortwave radiation (Iacono et al. 2008; Mlawer et
al. 1997), and ACM2 PBL (Pleim 2007) parameterization schemes on all three domains, along with the Kain-Fritsch cumulus
scheme (Kain 2004) on the outer two domains. The ACM2 PBL scheme is a hybrid local and non-local first-order closure
scheme that attempts to capture both subgrid and supergrid-scale fluxes (Pleim 2007). When conditions are stable, only the
local closure portion of the ACM2 scheme is used. Surface energy fluxes (sensible, latent, and ground) and changes in soil
moisture and soil temperature were simulated using the Pleim-Xu LSM (Gilliam and Pleim 2010; Xiu and Pleim, 2001).
Because this LSM only contains two layers (0-1 cm and 1-100 cm depth), indirect soil moisture and soil temperature nudging
is used to improve the accuracy of these variables. The indirect nudging uses the weighted differences between simulated 2-



m air temperature and relative humidity with available surface observations to reduce biases in the modeled soil moisture and soil temperature (Pleim and Gilliam 2009; Pleim and Xiu 2003). The 1-100 cm soil temperature was initialized as the average 2-meter temperature for the 10-day spin-up period (12-22 May 2017) using the IPXWRF utility (Pleim and Gilliam, 2009). In addition, analysis nudging was used to continuously adjust the temperature, water vapor, and winds above the PBL toward the 6-h GFS analyses (e.g., Borge et al. 2008; Campbell et al. 2018; Harkey and Holloway 2013; Otte 2008a, b; Otte et al. 2012; Pleim and Gilliam 2009). Finally, hourly surface observations of temperature, humidity, winds, and sea level pressure from the Meteorological Assimilation Data Ingest System (MADIS, https://madis.ncep.noaa.gov/) were used to perform surface nudging on all domains via the WRF OBSGRID utility.

A second simulation was performed using the YSU PBL (Hong et al. 2006), Noah LSM (Chen and Dudhia, 2001; Ek et al. 2003), and Thompson microphysics (Thompson et al. 2008, 2016) schemes, which is hereafter referred to as the "YNT" configuration. Like the EPA simulation, this configuration employed the RRTMG longwave and shortwave radiation and Kain-Fritsch cumulus schemes on the outer two domains, along with grid nudging toward the GFS temperature, humidity, and wind analyses above the PBL. This particular set of schemes was chosen based on our previous studies showing that they performed well during the warm season across the United States (e.g., Harkey and Holloway 2013; Cintineo et al. 2014; Greenwald et al. 2016; Griffin et al. 2021; Henderson et al. 2021). Because there are dozens of parameterization schemes to choose from in the WRF model, we do not aim to find necessarily the best physics suite but instead to assess the potential of using other schemes to improve upon the performance of the baseline EPA configuration. The YSU PBL scheme is a first-order, non-local closure scheme that allows non-local mixing with explicit entrainment processes at the top of the PBL (Hong et al. 2006; Hong 2010). The Noah LSM is a community model that has been widely used within the weather and climate modeling communities (Campbell et al. 2019). It contains four soil layers (0-10, 10-40, 40-100, and 100-200 cm depth) along with vegetation canopy, soil drainage, and runoff models that allow it to simulate surface hydrological and radiative processes. A realistic representation of land surface processes becomes increasingly important when moving towards higher model resolutions (e.g., Sutton et al. 2006; Case et al. 2008).

The remaining six simulations (Table 1) use the YNT configuration as their baseline. These simulations are designed to assess the impact of three high-resolution surface initialization datasets and analysis nudging above 2 km (rather than above the PBL) on the model accuracy when used individually or in combination. In particular, three simulations were run where the standard climatological or coarse-resolution surface initialization datasets were replaced by high-resolution, real-time datasets depicting lake surface temperatures, green vegetation fraction (GVF), and soil moisture / soil temperature across the study region. These surface datasets and the methods used to incorporate them into the WRF model simulations are described in the next section. Simulations employing these datasets are referred to as "YNT_SST", "YNT_GVF", and "YNT_SOIL", respectively. Another experiment was performed where analysis nudging was used above 2 km rather than above the PBL, which is referred to as the "YNT_N2KM" simulation. This change in nudging compared to the EPA and YNT baseline experiments was motivated by a modeling study by Odman et al. (2019) showing that the evolution of the nocturnal low-level jet across the Great Lakes region was more accurately simulated when nudging was withheld in the lower troposphere (e.g., below 2 km) when the PBL is shallow. Differences in the nocturnal low-level jet could affect the transport of ozone and its precursors from urban regions to Lake Michigan during the overnight hours. Finally, two "combination" simulations were performed where the 2-km analysis nudging approach was used along with all three of the high-resolution surface initialization datasets ("YNT_SSNG") or only with the lake surface temperature and soil datasets ("YNT_SSN"). The latter simulation is included because it was found that this combination of surface datasets and analysis nudging generally led to the best results.

**Table 1. List showing the parameterization schemes, model initialization datasets, and nudging approaches used during each of the eight WRF model experiments. Acronyms are described in the text.**

|  | EPA | YNT | YNT_SST | YNT_GVF | YNT_SOIL | YNT_N2KM | YNT_SSNG | YNT_SSN |
|---|---|---|---|---|---|---|---|---|
| PBL | ACM2 | YSU | YSU | YSU | YSU | YSU | YSU | YSU |
| LSM | Pleim-Xu | Noah | Noah | Noah | Noah | Noah | Noah | Noah |



| Surface Layer | Pleim-Xu | Monin-Obukhov | Monin-Obukhov | Monin-Obukhov | Monin-Obukhov | Monin-Obukhov | Monin-Obukhov | Monin-Obukhov |
|---|---|---|---|---|---|---|---|---|
| Micro. | Morrison | Thompson | Thompson | Thompson | Thompson | Thompson | Thompson | Thompson |
| Cumulus | Kain-Fritsch | Kain-Fritsch | Kain-Fritsch | Kain-Fritsch | Kain-Fritsch | Kain-Fritsch | Kain-Fritsch | Kain-Fritsch |
| IC / LC | GFS-FNL | GFS-FNL | GFS-FNL | GFS-FNL | GFS-FNL | GFS-FNL | GFS-FNL | GFS-FNL |
| SST | *default* | *default* | GLSEA | *default* | *default* | *default* | GLSEA | GLSEA |
| GVF | *default* | *default* | *default* | VIIRS | *default* | *default* | VIIRS | *default* |
| Soil | Initialized as 10-day ave. of 2-m temperature | *default* | *default* | *default* | SPoRT LIS | *default* | SPoRT LIS | SPoRT LIS |
| Nudging | analysis above the PBL; obs nudging to MADIS | analysis, above PBL | analysis, above PBL | analysis, above PBL | analysis, above PBL | analysis, above 2 km | analysis, above 2 km | analysis, above 2 km |

## 2.2 Surface initialization datasets

### 2.2.1 Lake surface temperatures

Daily maps of Great Lakes surface temperatures, with a horizontal resolution of ~1.3 km, were obtained from the Great Lakes Surface Environmental Analysis (GLSEA) produced at the NOAA Great Lakes Environmental Research Laboratory (Schwab 1992). The lake surface temperatures are estimated using clear-sky infrared brightness temperatures from the Advanced Very High-Resolution Radiometer onboard multiple polar-orbiting satellites. If a surface retrieval is not possible during a given day due to persistent cloud cover, a smoothing algorithm is applied to the previous analysis to maintain complete coverage. Only satellite observations are used to produce the daily lake surface temperature analyses, which were then used to overwrite the simulated surface temperatures for Great Lakes grid points at 00 UTC each day in the YNT_SST, YNT_SSN, and YNT_SSNG simulations. Replacing the coarse-resolution (0.25°) GFS FNL surface temperatures (Fig. 2a) with the GLSEA analyses (Fig. 2b) led to warmer lake temperatures near the shoreline, especially along northern parts of Lake Michigan where temperatures were > 2 K warmer, and cooler temperatures across the rest of the lake, when averaged over the 22 May – 22 June 2017 time period (Fig. 2c). This spatial pattern indicates that the finer horizontal resolution of the GLSEA dataset allows it to capture warmer temperatures in shallower waters near the shoreline while also depicting the cooler mid-lake temperatures due to the cooler-than-normal weather conditions that prevailed across the region in May (NCEI 2017).

### 2.2.2 VIIRS green vegetation fraction

GVF is the photosynthetically active fractional green vegetation cover within a grid cell, with higher values indicating more extensive actively transpiring vegetation. It is a key parameter in an LSM because vegetation representation is used to partition the incoming solar radiation into sensible, latent, and ground heat fluxes, where the latent heat flux is largely due to vegetation transpiration (e.g., Yin et al. 2016). Surface latent heat flux is sensitive to GVF because vegetation roots are able to access deeper soil moisture that would not otherwise be able to evaporate (Miller et al. 2006). For this study, we used daily global GVF derived using observations from the Visible Infrared Imaging Radiometer Suite (VIIRS; Vargas et al. 2015) in place of the default monthly climatology to constrain the evolution of vegetation in the YNT_GVF and YNT_SSNG simulations. The VIIRS GVF composite product is generated daily at 4-km resolution and available from the NOAA Comprehensive Large Array-data Stewardship System (CLASS). The real-time daily GVF analyses were used to overwrite the default monthly



climatological vegetation fraction data used by the WRF model at 00 UTC each day. Using real-time, satellite derived GVF in place of a monthly GVF climatology has been shown to improve the representation of the surface energy budget and subsequent model forecasts during the warm season (Case et al. 2014). In Fig. 2f, it is evident that use of the real-time GVF led to lower leaf area index (Fig. 2e; computed internally by the WRF model) across most of the domain compared to the climatological vegetation data (Fig. 2d), with the exception of some forested regions in the northern portion of the domain and bands of enhanced leaf area index surrounding metropolitan areas such as Chicago. The lower leaf area index in agricultural areas is consistent with delayed crop growth due to the cool spring weather, whereas the bands of higher leaf area index represent the impact of urban sprawl since the climatological vegetation data shown in Fig. 2d was generated using satellite observations from the late 1980s and early 1990s (see Gutman et al. 1995).



**Figure 2.** Average lake surface temperatures from the (a) YNT and (b) YNT_SST simulations, with their differences shown in (c). Average leaf area index from the (d) YNT and (e) YNT_GVF simulations, with their differences shown in (f). Average 0-10 cm soil temperatures from the (g) YNT and (h) YNT_SOIL simulations, with their differences shown in (i). Average 0-10 cm soil moisture content from the (j) YNT and (k) YNT_SOIL simulations, with their differences shown in (l). The averages for each variable were computed using data valid at 00 UTC each day during the 22 May – 22 June 2017 time period.



**2.2.3 SPoRT LIS soil moisture and temperature analyses**

A customized version of the Land Information System (LIS; Kumar et al. 2006) run at the Short-term Prediction Research and Transition Center (SPoRT) was used to generate high-resolution soil moisture and soil temperature analyses. Version 3.6 of the Noah LSM (Chen and Dudhia 2001) was run on a 1-km resolution domain covering the central and eastern United States and nearby portions of southern Canada. Required inputs to run the Noah LSM were obtained from hourly analyses of surface pressure, 2-m temperature, 2-m specific humidity, 10-m wind speed, and downwelling shortwave and longwave radiation from the North American Land Data Assimilation System – Phase 2 (NLDAS-2; Xia et al. 2012). Quantitative precipitation estimates (QPE) were obtained from the Multi-Radar Multi-Sensor (MRMS) gauge-adjusted radar product (Zhang et al. 2016), the Global Data Assimilation System (GDAS; Wang et al. 2013), and NLDAS-2. A simple blending methodology was used to incorporate the multiple sources of QPE because evaluation of the real-time SPoRT-LIS product (Case 2016; Case and Zavodsky 2018; Blankenship et al. 2018) and preliminary LIS experiments during this study revealed that the NLDAS-2 and MRMS precipitation products have a dry bias across the region. To reduce this bias, the precipitation forcing used the average of the highest two values of the MRMS, GDAS, and NLDAS-2 QPE datasets. Inspection of the blended precipitation product showed that the precipitation bias was reduced, while preserving small-scale spatial details in the MRMS QPE product. Daily VIIRS GVF composites were also used to constrain vegetation during the offline LIS-Noah simulation.

Following an initial spin-up of LIS using NLDAS-2 forcing data from 2012-2016 to remove memory of the prescribed initial conditions, the final analysis from this run was used to restart the simulation on 01 January 2012 using NLDAS-2 atmospheric forcing data, VIIRS GVF, and the merged QPE product. Soil moisture and soil temperature analyses from this LIS simulation were then used to replace the corresponding variables in the YNT_SOIL, YNT_SSN, and YNT_SSNG simulations at 00 UTC each day from 12 May – 22 June 2017. Direct insertion into the WRF model was possible because of the similarly configured Noah LSM used in both the LIS and WRF simulations. Comparison of the 0-10 cm soil temperatures from the GFS (Fig. 2g) and LIS (Fig. 2h), averaged over the 22 May – 22 June 2017 period, shows that the topsoil temperatures are noticeably cooler in the LIS data across most of the region, except for northern parts of Wisconsin and Michigan. The cooler temperatures are most prominent in suburban regions where the largest increases in GVF also occurred (Fig. 2f). For 0-10 cm soil moisture, the LIS analyses are generally wetter across the domain (Fig. 2l), with the largest increases across forested regions of Wisconsin and Michigan. Deeper soil layers exhibited similar differences between the GFS FNL and LIS datasets (not shown).

**2.3 Analysis methods**

The accuracy of the WRF model simulations was assessed using hourly surface observations of temperature, humidity, and winds from MADIS during 22 May – 22 June 2017. Note that these surface observations were also used to perform surface nudging during the EPA simulation, which will impact the results presented in Section 3 because surface nudging was not used during any of the YNT simulations. The model evaluations are performed on all three domains using observations from stations located on the innermost domain surrounding Lake Michigan, which allows us to assess the behavior of each configuration as a function of spatial resolution using the same set of stations. Version 1.4 of the Atmospheric Model Evaluation Tool (AMET; Appel et al. 2011) from the EPA was used to collocate hourly observed and modeled values in a grid cell where a particular observation station was located; and to calculate model performance statistics including bias and root mean square error.

**3. Results**

**3.1 Assessment of EPA and YNT baseline experiments**

This section contains a high-level assessment of the accuracy of the EPA and YNT baseline experiments on each domain, with a more detailed evaluation of all experiments on the 1.3-km resolution domain provided in Section 3.2. Figure 3 shows 2-m temperature, 2-m water vapor mixing ratio, and 10-m wind speed errors for each domain computed using hourly surface observations. The left column shows the bias for each variable and experiment, whereas the center and right columns show





the percentage changes in RMSE for each experiment relative to the EPA and YNT baseline experiments, respectively. A negative (positive) value for a given variable and domain indicates that the RMSE for that experiment is smaller (larger) than the actual RMSE for the corresponding baseline experiment plotted in the gray box.

| | Bias | | | % RMSE Change vs. EPA | | | % RMSE Change vs. YNT | | |
|---|---|---|---|---|---|---|---|---|---|
| **a) 2-m Temperature [K]** | | | | **b) 2-m Temperature [K]** | | | **c) 2-m Temperature [K]** | | |
| Simulation | 12 km | 4 km | 1.3 km | 12 km | 4 km | 1.3 km | 12 km | 4 km | 1.3 km |
| EPA | -0.12 | -0.40 | 0.16 | 2.03 | 2.23 | 3.00 | | | |
| YNT | 0.16 | 0.47 | 0.55 | 13.08 | 0.45 | -25.18 | 2.30 | 2.24 | 2.25 |
| YNT_SST | 0.17 | 0.48 | 0.56 | 12.44 | -0.13 | -25.58 | -0.57 | -0.58 | -0.53 |
| YNT_SOIL | -0.39 | -0.19 | -0.22 | 11.95 | -4.62 | -30.41 | -1.00 | -5.04 | -6.99 |
| YNT_N2KM | 0.25 | 0.58 | 0.67 | 12.44 | -0.18 | -24.68 | -0.57 | -0.62 | 0.67 |
| YNT_GVF | -0.28 | -0.02 | -0.03 | 12.54 | -1.88 | -27.91 | -0.48 | -2.32 | -3.65 |
| YNT_SSNG | -0.56 | -0.32 | -0.38 | 10.62 | -5.20 | -29.71 | -2.17 | -5.62 | -6.06 |
| YNT_SSN | -0.29 | -0.07 | -0.09 | 9.00 | -7.44 | -31.91 | -3.61 | -7.85 | -8.99 |
| **d) 2-m Mixing Ratio [g/kg]** | | | | **e) 2-m Mixing Ratio [g/kg]** | | | **f) 2-m Mixing Ratio [g/kg]** | | |
| Simulation | 12 km | 4 km | 1.3 km | 12 km | 4 km | 1.3 km | 12 km | 4 km | 1.3 km |
| EPA | 0.91 | 1.28 | 1.35 | 1.85 | 2.03 | 2.07 | | | |
| YNT | 0.19 | 0.00 | -0.20 | -19.96 | -28.69 | -29.85 | 1.48 | 1.44 | 1.45 |
| YNT_SST | 0.20 | 0.00 | -0.20 | -20.66 | -29.19 | -30.48 | -0.88 | -0.69 | -0.90 |
| YNT_SOIL | 0.24 | 0.10 | -0.02 | -20.12 | -29.14 | -31.40 | -0.20 | -0.62 | -2.21 |
| YNT_N2KM | 0.22 | 0.05 | -0.14 | -19.69 | -28.10 | -28.88 | 0.34 | 0.83 | 1.38 |
| YNT_GVF | 0.30 | 0.17 | 0.02 | -20.12 | -28.69 | -30.77 | -0.20 | 0.00 | -1.31 |
| YNT_SSNG | 0.36 | 0.28 | 0.24 | -22.33 | -29.83 | -31.54 | -2.96 | -1.59 | -2.41 |
| YNT_SSN | 0.27 | 0.14 | 0.04 | -21.20 | -29.83 | -32.03 | -1.55 | -1.59 | -3.10 |
| **g) 10-m Wind Speed [m/s]** | | | | **h) 10-m Wind Speed [m/s]** | | | **i) 10-m Wind Speed [m/s]** | | |
| Simulation | 12 km | 4 km | 1.3 km | 12 km | 4 km | 1.3 km | 12 km | 4 km | 1.3 km |
| EPA | 0.05 | -0.17 | -0.14 | 1.52 | 1.51 | 1.63 | | | |
| YNT | 0.45 | 0.34 | 0.36 | 6.26 | 2.19 | -3.32 | 1.61 | 1.54 | 1.57 |
| YNT_SST | 0.46 | 0.34 | 0.36 | 6.52 | 2.52 | -2.40 | 0.25 | 0.32 | 0.95 |
| YNT_SOIL | 0.38 | 0.24 | 0.23 | 5.07 | 1.26 | -4.49 | -1.12 | -0.91 | -1.21 |
| YNT_N2KM | 0.42 | 0.32 | 0.34 | 4.61 | 0.60 | -5.05 | -1.55 | -1.56 | -1.78 |
| YNT_GVF | 0.60 | 0.54 | 0.60 | 10.87 | 7.97 | 4.06 | 4.34 | 5.65 | 7.64 |
| YNT_SSNG | 0.53 | 0.47 | 0.49 | 8.04 | 5.25 | -0.25 | 1.67 | 2.99 | 3.18 |
| YNT_SSN | 0.36 | 0.23 | 0.22 | 3.82 | -0.20 | -6.52 | -2.29 | -2.34 | -3.31 |

-1     0     1       -40     0     40       -10     0     10

**Figure 3.** Summary statistics showing the (a) 2-m temperature bias for each experiment, along with the percentage change in the 2-m temperature root mean square error (RMSE) for a subset of experiments relative to the (b) EPA baseline and (c) YNT baseline experiments, respectively. Statistics for the 12-km, 4-km, and 1.3-km resolution domains were computed using hourly data from all stations located on the 1.3-km resolution domain during 22 May – 22 June 2017. The actual RMSEs for the baseline experiments (gray boxes) are also shown. Blue (orange) shading indicates a negative (positive) bias for a given experiment in (a), whereas blue (orange) shading depicts smaller (larger) RMSE in a given experiment relative to the EPA and YNT baseline experiments in (b) and (c). (d-f) Same as (a-c), except for showing statistics for 2-m mixing ratio. (g-i) Same as (a-c), except for showing statistics for 10-m wind speed.




Inspection of the YNT statistics reveals a consistent pattern in the RMSE where the percentage changes for each variable either
switch from positive to negative, or become more strongly negative, as the model resolution increases from 12 km to 1.3 km.
For temperature, the RMSE improves from being 13.08% larger than the EPA on the 12-km domain to 25.18% smaller on the
1.3-km domain (Fig. 3b). A similar pattern is present for 10-m wind speed where the RMSE is 6.26% larger on the 12-km
domain, but then steadily decreases so that the RMSE becomes 3.32% smaller on the 1.3-km domain (Fig. 3h). Though the
EPA simulation has much larger bias and RMSE for 2-m mixing ratio on all domains (Fig. 3d, 3e), the same pattern emerges
with this variable where it becomes less accurate at higher resolutions. Aside from using different parameterization schemes,
the only difference between the baseline experiments is the use of soil and surface observation nudging in the EPA simulation.
These results indicate that the EPA physics suite becomes less accurate, or the soil and surface nudging methods become less
effective, at higher model resolutions. Because surface nudging is used on all domains during the EPA simulation, the poor
performance on the 1.3-km domain suggests that it is no longer able to overcome deficiencies in the parameterization schemes,
especially the Pleim-Xu LSM (see Section 3.3), at higher spatial resolutions. It is also possible that the lack of dense surface
observations makes it challenging to effectively apply surface nudging at high resolutions since the observations lack sufficient
spatial detail to capture small-scale atmospheric and land surface features. Regardless, Fig. 3 shows that the YNT configuration
provides superior performance on the 1.3-km domain when averaged across all stations. In the following sections, we will use
results from this domain to examine the impacts of the surface initialization datasets and analysis nudging on the model
accuracy with respect to both the EPA and YNT baseline experiments.

**3.2 YNT sensitivity experiments**

**3.2.1 2-m temperature analysis**

To examine regional differences in model performance, Fig. 4 shows the 2-m temperature bias and RMSE computed separately
for each station using hourly observations from 22 May – 22 June 2017. For the EPA simulation, there is a north-south gradient
in the RMSE, with the largest errors across northern Illinois and Indiana (Fig. 4a). Stations near Lake Michigan generally have
the smallest RMSE due to its moderating influence on local weather conditions. Similar to the RMSE, the smallest biases
occurred in the northern part of the domain and along the eastern shoreline; however, biases along the western shoreline are
larger and of comparable magnitude to those at inland locations across Wisconsin and Illinois. Overall, the EPA simulation
had an RMSE of 3 K and a bias of 0.16 K when averaged across all stations (Figs. 3a-b). Switching to the YNT parameterization
suite greatly reduced the RMSE by 25.18% across the entire domain (Fig. 3b); however, the bias increased to 0.55 K (Fig. 3a).
The largest RMSE reductions (up to 45%) occurred in rural areas of northern Illinois, with similar RMSEs found across the
entire domain (Fig. 4b). The larger positive temperature bias in the YNT baseline simulation is primarily due to larger errors
in Wisconsin and within densely populated urban areas along the western Lake Michigan shoreline from Chicago to
Milwaukee (Fig. 4f). A mixed pattern of larger and smaller biases occurred elsewhere across the domain.
Inspection of the YNT sensitivity experiments shows that the smallest RMSEs occurred during the YNT_SOIL, YNT_SSN,
and YNT_SSNG simulations, with the average RMSE reduced by 29.7% to 31.9% relative to the EPA baseline (Fig. 3b) and
from 6.0% to 9.0% relative to the already greatly improved YNT baseline (Fig. 3c). On an individual basis, the high-resolution
soil initialization dataset (YNT_SOIL) had the largest positive impact at most stations (Fig. 4d), whereas slightly larger RMSEs
were observed when using nudging (YNT_N2KM) (Fig. 4j). Comparison of the YNT_SSN and YNT_SSNG simulations (Fig.
4l, 4p) shows that inclusion of the VIIRS GVF initialization dataset during the YNT_SSNG simulation led to slightly larger
RMSE for stations near the lakeshore, but similar or smaller errors for stations located further inland.
The bias pattern for the YNT simulations is more complex. Overall, the bias was largest (0.67 K) in the YNT_N2KM
simulation, with the smallest biases occurring in the YNT_GVF (-0.03 K) and YNT_SSN (-0.09 K) simulations (Fig. 3a).
Switching from the EPA to YNT baseline configurations led to larger biases across most of the domain, especially along the
southwestern shoreline of Lake Michigan (Fig. 4e-f). The high-resolution SST dataset had a minimal impact on the biases
(Fig. 4g) whereas they were smaller in the YNT_SOIL (Fig. 4h) and YNT_GVF (Fig. 4m) simulations relative to the YNT
baseline. Use of these two land datasets however led to much larger negative biases along the eastern shoreline of Lake



Michigan. When 2-km analysis nudging was used (YNT_N2KM), larger positive biases occurred from Chicago to Milwaukee,
with smaller biases along the eastern shoreline (Fig. 4n). The increased RMSE and bias near the western shoreline compared
to locations further inland during the YNT_N2KM simulation suggests that the modified nudging routine (applied to heights
above 2 km instead of above the PBL) may not work well for areas near Lake Michigan due to the moderating influence of
the lake on the PBL. Because the PBL tends to be more stable and shallower for locations over and near Lake Michigan due
to the cooler surface temperatures, this means that confining analysis nudging to above 2 km limits its ability to constrain the
evolution of the lower troposphere during the YNT_N2KM simulation.

### 3.2.2 2-m water vapor analysis

For the 2-m water vapor mixing ratio, switching to the YNT physics suite led to nearly a 30% reduction in the station-average
RMSE during the YNT simulation relative to the EPA baseline (Fig. 3e), with additional incremental reductions occurring in
all sensitivity experiments except for YNT_N2KM (Fig. 3f). The much lower RMSE in all of the YNT simulations is primarily
due to the notable reduction in bias (Fig. 3d). Whereas the EPA configuration had a large moist bias (1.35 g kg$^{-1}$), the YNT
bias was much smaller and also became negative (-0.20 g kg$^{-1}$). The bias was further reduced during most of the sensitivity
experiments, with only a slight increase during the YNT_SSNG simulation. Overall, the YNT_SSN simulation had the smallest
RMSE and a bias close to zero when averaged across all of the stations.

Looking more closely at the individual stations (Fig. 5), it is evident that almost all of them have a positive (e.g., moist) bias
when the EPA configuration is used (Fig. 5e). The largest biases are located in the southern portion of the domain, especially
for stations near the lakeshore. In contrast, about two-thirds of the stations exhibit a negative bias during the YNT simulation
(Fig. 5f). The spatial pattern of the biases is similar during all of the YNT sensitivity experiments; however, their magnitude
is generally smaller, which is consistent with the overall statistics (Fig. 3d). For RMSE, the largest errors in the EPA simulation
occur primarily along the southern end of Lake Michigan, with generally smaller errors in the northern half of the domain
(Fig. 5a). The RMSE during the YNT simulation is smaller at most locations, especially along the shoreline, though a few
stations near the western shoreline have larger errors (Fig. 5b). Use of the SOIL and GVF initialization datasets reduced the
errors at these nearshore locations (Fig. 5d, 5i), with the smallest errors at most stations occurring during the combination
experiments (YNT_SSN and YNT_SSNG). As was the case with 2-m temperature, the most accurate 2-m water vapor analyses
were obtained during the YNT_SSN simulation.

### 3.2.3 10-m wind speed analysis

Compared to the temperature and water vapor fields, changes to the 10-m wind speed statistics were much more modest during
the YNT simulations. Switching from the EPA configuration to the YNT configuration led to a 3.32% reduction in the RMSE,
but a larger bias that also changed sign from negative to positive (Fig. 3g). For the YNT experiments, the average RMSE was
slightly smaller during the YNT_SOIL and YNT_N2KM simulations (-1.21% and -1.78%, respectively), but slightly larger
(0.95%) during the YNT_SST simulation compared to the YNT baseline (Fig. 3i). Use of the GVF surface initialization dataset
led to a 7.64% increase in the RMSE during the YNT_GVF simulation, primarily due to a larger wind speed bias. Overall, the
most accurate wind speed analyses were achieved during the YNT_SSN simulation, with an RMSE reduction of 6.52% across
all stations.

Spatially, there is a latitudinal gradient in wind speed errors during the EPA simulation. The largest RMSEs are located across
the southern part of the domain (Fig. 6a), with mostly negative wind speed biases (up to 2 m s$^{-1}$) in the same region transitioning
to a mix of negative and positive biases in northern Wisconsin and Michigan (Fig. 6e). The RMSE and bias were much smaller
for stations around the southern shoreline of Lake Michigan during the YNT simulation; however, slightly larger RMSEs are
present across inland locations in the northern part of the domain (Fig. 6b). A similar spatial pattern of changes relative to the
EPA baseline occurred during the YNT sensitivity experiments, though the errors are generally larger during the YNT_GVF
simulation (Fig. 6i, 6m) and smaller during the YNT_SOIL (Fig. 6d, 6h) and YNT_N2KM (Fig. 6j, 6n) simulations. The poor
performance of the YNT_GVF and YNT_SSNG simulations is primarily due to larger errors across inland areas of Wisconsin
where there are large positive wind speed biases (Fig. 6m, 6p), with similar errors elsewhere in the domain.



**Figure 4.** Maps showing the 2-m temperature root mean square error (RMSE) and bias for each station on the 1.3-km domain computed using hourly data from 22 May – 22 June 2017. Statistics for the EPA, YNT, YNT_SST, and YNT_SOIL experiments are shown in (a)–(h), whereas results for the YNT_GVF, YNT_N2KM, YNT_SSN, and YNT_SSNG experiments are shown in (i)–(p).





## 2-m Mixing Ratio

Figure 5. Same as Fig. 4, except for 2-m water vapor mixing ratio.





**Figure 6.** Same as Fig. 4, except for 10-m wind speed.



### 3.2.4 Diurnal error characteristics

Fig. 7 shows the diurnal evolution of RMSE and bias for 2-m temperature, 2-m water vapor mixing ratio, and 10-m wind speed at hourly intervals starting at 1900 local standard time (LST). The time series were computed by averaging over data from all stations on the 1.3-km domain. Overall, it is apparent that the EPA simulation contains very different diurnal error patterns than the YNT simulations. For example, the 2-m temperature bias exhibits a prominent diurnal cycle (Fig. 7b) characterized by large positive/warm (negative/cool) biases during the night (day), resulting in an overall damping of the diurnal temperature cycle. The warm biases exceed 2.5 K during most of the night (23 – 05 LST) and the cold biases are < -2 K for several hours during the daytime (1000–1300 LST). These results indicate that the small temperature bias in the summary statistics for the EPA simulation (Fig. 3a) is misleading because it obscures the presence of substantial biases of opposite signs during the day and night. The RMSE is also much larger during the EPA simulation (Fig. 7a), with local maxima of 4.4 K and 3.1 K at 0000 and 1200 LST, respectively, corresponding to peaks in the biases. Switching to the YNT greatly reduces the temperature RMSE, and the bias time series is no longer characterized by the highly amplified diurnal pattern seen in the EPA simulation. Examination of the YNT sensitivity experiments shows similar error patterns to the YNT baseline. The largest differences occur at night when use of the GVF and SOIL datasets leads to smaller biases. In contrast, confining the analysis nudging to above 2 km AGL (YNT_N2KM) slightly increases the RMSE and bias during the nighttime relative to the YNT baseline.

For water vapor, the EPA simulation again exhibits much larger bias and RMSE than the other simulations (Fig. 7c, 7d). It has a large moist bias that ranges from 0.9 g kg$^{-1}$ shortly after sunrise to 1.7 g kg$^{-1}$ near 1900 LST, before decreasing to a relatively stable bias of 1.3 g kg$^{-1}$ during the night. The RMSE is much smaller in the YNT baseline simulation, with a dry bias evident for all but the evening hours (1900-2200 LST). As is the case for temperature, the RMSE is smallest during the late-night hours and then steadily increases during the day before reaching its maximum in the evening. All of the YNT sensitivity experiments have similar RMSE and bias patterns to the YNT baseline, with the smallest (largest) spread between simulations occurring during the nighttime (daytime) hours, possibly due to differences in the PBL depth and surface energy balance (see Fig. 8). Comparison of the 10-m wind speed time series reveals that the EPA simulation has the smallest bias (~ 0.35 m s$^{-1}$) during the night, but that the wind speeds are weaker than observed during the daytime, with the largest biases (-0.8 m s$^{-1}$) occurring at noon (Fig. 7f). This diurnal pattern in the EPA simulation, characterized by winds that are too strong (weak) during the night (day), stands in contrast to the mostly positive biases in the YNT simulations. The biases are tightly clustered in all of the YNT experiments during the nighttime hours (2200–0700 LST), with the exception of the two simulations employing the GVF initialization dataset (YNT_GVF and YNT_SSNG) that are characterized by persistently larger positive biases. These two simulations also have the largest RMSE (Fig. 7e). Further research is necessary to determine why incorporation of the high-resolution GVF dataset leads to larger surface wind speed errors.







By

**Figure 7.** Time series showing the diurnal evolution of (a-b) 2-m temperature root mean square error (RMSE) and bias, (c-d) 2-m water vapor mixing ratio RMSE and bias, and (e-f) 10-m wind speed RMSE and bias at hourly intervals starting at 1900 local standard time (LST). Errors were computed for each model simulation using observations from all stations located on the 1.3-km resolution domain during 22 May – 22 June 2017.



### 3.2.5 Surface Energy Budget Considerations

Near-surface atmospheric conditions can be strongly impacted by the partitioning of net surface radiation into sensible, latent, and ground heat fluxes (Santanello et al. 2018). To examine this more closely, Fig. 8 shows time series depicting the average diurnal evolution of the PBL height, net surface radiation, and sensible, latent, and ground heat fluxes during 22 May – 22 June 2017 computed using data from stations on the 1.3-km domain to maintain consistency with earlier results. Because in-situ flux and PBL height observations are not available across the entire domain, the aim is not to examine the accuracy of the simulated surface energy fluxes and PBL height, but rather to use these variables to help explain differences in the near-surface temperature, water vapor, and wind speed errors in the model simulations. All of the variables were obtained directly from the WRF output files. The net surface radiation is defined as the sum of the sensible, latent, and ground heat fluxes.

Inspection of Fig. 8 reveals large differences between the EPA and YNT simulations. The PBL is ~100-200 m deeper in the EPA simulation during the nighttime but then becomes much shallower than the YNT simulations from mid-morning through the afternoon (1000–1600 LST) with the daytime maximum in PBL height occurring 1-2 h later (Fig. 8a). The EPA simulation is also characterized by a smoother and less amplified diurnal evolution. For the YNT simulations, the PBL heights are tightly clustered during the night (2100 – 0700 LST) but begin to diverge during the morning and reach their largest differences during the afternoon. In particular, simulations employing the high-resolution soil moisture analyses (YNT_SOIL, YNT_SSN, and YNT_SSNG) have average PBL heights that are ~100 m lower than the other YNT simulations. These three simulations also have slightly lower sensible heat flux (Fig. 8c) and higher latent heat flux during the afternoon (Fig. 8d), which is consistent with the wetter and cooler topsoil layer in the SPoRT LIS analyses (Fig. 2g-l) and cooler 2-m temperatures (Figs. 3a, 7b). Using the SST and GVF datasets and confining analysis nudging to above 2 km had minimal impact on the PBL heights in the YNT_SST, YNT_GVF, and YNT_N2KM simulations; however, sensible and latent heat fluxes are slightly smaller during the afternoon in the YNT_GVF simulation.

Comparison of the EPA and YNT simulations also reveals large differences in the surface energy flux time series. For example, the EPA simulation has much smaller sensible heat flux during the daytime (Fig. 8c) and the latent heat flux remains relatively large during the night (Fig. 8d). Though the EPA and YNT simulations produce similar magnitudes of latent heat flux during the day, the afternoon maximum is delayed by 2 h in the EPA simulation. The combination of a shallower PBL during the day (Fig. 8a) and higher latent heat flux at night likely contributes to the persistent large moist bias in the 10-m water vapor mixing ratio (Figs. 3d, 7d) during the EPA simulation. Another noteworthy feature of the EPA simulation is that the ground heat flux remains negative at all times. This unphysical behavior stands in sharp contrast to the more realistic evolution during the YNT simulations where the positive (negative) ground heat flux during the night (day) indicates that heat is being transferred from (toward) the ground toward (from) the atmosphere due to cooler (warmer) surface temperatures. These results indicate that the poor performance of the EPA simulation on the 1.3-km domain when assessed using near-surface moisture, temperature, and wind observations is likely due to the presence of vastly different and sometimes unphysical surface energy fluxes.

The lower accuracy of the EPA simulation on the 1.3-km domain could be due to the use of soil nudging in the Pleim-Xu LSM because the observations used in the nudging approach are typically too coarse to provide the fine-scale geographically induced details needed to perform high-quality soil nudging (J. Pleim, personal comm.). This possibility is supported by Fig. 9, which shows the evolution of the PBL height and surface fluxes on the 12-km domain computed using simulated data from all stations on the 1.3-km domain. Differences between the EPA and YNT simulations are much smaller both in timing and magnitude on the 12-km domain. For example, the time series for PBL height, sensible heat flux, and latent heat flux are very similar for all of the simulations. Though the ground heat flux time series for the EPA simulation continues to be an outlier at this resolution, it now has the correct diurnal cycle with positive (negative) values during the night (day). The improved simulation of surface fluxes on the 12-km domain likely contributes to the more accurate temperature and wind speed analyses in the EPA simulation at that resolution (Fig. 3a-b, 3g-h). The presence of persistently higher latent heat flux (Fig. 9d) leads to a positive moisture bias in the EPA simulation (Fig. 3d-e); however, the bias is smaller on the 12-km domain than it was on the 1.3-km domain. Inspection of each of the surface energy fluxes and PBL height on the 4-km domain revealed larger differences between the EPA and YNT simulations (not shown), but not as large as those on the 1.3-km domain. Together, these results show that the EPA simulation performs well at 12-km resolution, but that its accuracy decreases with increasing model resolution.




**Figure 8. Time series showing the diurnal evolution of the (a) planetary boundary layer height, (b) net radiation, (c) sensible heat flux, (d) latent heat flux, and (e) ground heat flux at hourly intervals starting at 1900 local standard time (LST), averaged over all stations on the 1.3-km domain during 22 May – 22 June 2017. Results are shown individually for each of the model simulations.**





**Figure 9. Same as Fig. 8, except for showing results on the 12-km domain. Time series were computed using simulated data from all**
**stations located on the 1.3-km domain.**



## 4. Discussion and conclusions

In this study, eight WRF model simulations were performed to assess the impact of different parameterization schemes, surface initialization datasets, and analysis nudging on the simulation of surface energy fluxes and near-surface atmospheric conditions in the Lake Michigan region during a 1-month period (22 May – 22 June 2017) corresponding to the LMOS field campaign. The simulations employed a triple-nested domain configuration containing 12-, 4-, and 1.3-km resolution grids, respectively. The "EPA" baseline simulation employed parameterization schemes and a model configuration similar to that used at the EPA, including soil and surface observation nudging. A second simulation ("YNT") was performed using different parameterization schemes that are easier to use because they do not require soil and surface observation nudging. Another important difference is that the YNT simulation used the more sophisticated Noah LSM to simulate land processes rather than the Pleim-Xu LSM that was used in the EPA simulation. The YNT configuration then served as the baseline to perform six additional simulations to assess the impact of three satellite- and model-derived surface initialization datasets and analysis nudging. Simulations were run where standard climatological or coarse-resolution surface initialization datasets were replaced by high-resolution, real-time datasets depicting lake surface temperatures, GVF, and soil moisture/soil temperature. Near-surface temperature, water vapor, and wind observations were used to assess the accuracy of each model simulation.

The EPA configuration generally produced more accurate analyses on the 12-km domain, with the exception of a large moist bias in the 2-m water vapor mixing ratio, but its accuracy greatly decreased with finer model grid resolution. The superior performance of the EPA simulation on the 12-km domain is partially an artifact of its use of surface observation nudging because the same observations used in the nudging routine were also used for verification. However, surface observation nudging was also used on the 4-km and 1.3-km domains in the EPA simulation, which indicates that it becomes less effective at constraining the evolution of the atmosphere at higher spatial resolutions. This is possible because the surface observations lack sufficient spatial density to accurately capture and constrain small-scale features associated with abrupt changes in land surface characteristics such as occurs along coastlines or the interface between urban and rural areas.

Evaluation of the EPA simulation showed that the diurnal evolution of the sensible and latent heat fluxes was similar to the YNT simulation on the 12-km domain but differed greatly on the 1.3-km nested domain where it had much smaller sensible heat flux during the daytime and larger latent heat flux at night. The increased latent heat flux combined with a shallower PBL contributed to the large moist bias in the 2-m water vapor mixing ratio. The evolution of the EPA ground heat flux was physically unrealistic on the 1.3-km domain because it remained negative at all times rather than changing signs between day and night as occurred during the YNT simulations. Because the evolution of the surface energy fluxes was more realistic on the 12-km domain, the poorer performance on the 4- and 1.3-km domains suggests that the Pleim-Xu LSM is unable to adequately represent surface fluxes at higher resolutions. This could be due to its use of two soil layers including a very shallow (1 cm) topsoil layer that make it difficult to fully represent fine-scale features and soil heat fluxes. Increasing the number of soil layers in the Pleim-Xu LSM could potentially improve its ability to simulate energy fluxes on high-resolution domains and reduce its dependence on nudging to constrain its evolution.

Inspection of the YNT statistics revealed a consistent pattern where the percentage change in the RMSEs for 2-m temperature, 2-m water vapor mixing ratio, and 10-m wind speed relative to the EPA baseline improved as the model resolution increased from 12 km to 1.3 km. The superior performance at higher resolutions when using the YNT configuration was achieved without using soil nudging or surface observation nudging. Switching to the YNT configuration led to substantial decreases in RMSE for 2-m temperature (25%) and 2-m water vapor mixing ratio (30%), and a more modest 3.3% reduction in the RMSE for 10-m wind speed, when assessed using all stations on the 1.3-km domain. Despite the already large error reductions when using the YNT parameterization suite, additional improvements occurred in most of the variables when the high-resolution surface initialization datasets were incorporated into the modeling platform. Evaluation of the YNT sensitivity experiments showed that the high-resolution soil initialization dataset had the largest positive impact on temperature and water vapor errors and the second largest impact on wind speed. Use of the GVF and SST datasets also led to more accurate temperature and water vapor simulations, but some degradations in the wind speed, especially when using the GVF dataset. Only the simulation employing analysis nudging above 2 km produced more accurate 10-m wind speed analyses; however, 2-m temperature errors were larger along the western shoreline of Lake Michigan when the nudging was confined to levels above 2 km instead of above the PBL.



This suggests that the modified nudging approach may not work well for areas near Lake Michigan where the PBL tends to be shallower because it reduces its ability to constrain the evolution of the lower troposphere. Despite this limitation, the most accurate near-surface simulations were obtained during the experiment that employed analysis nudging above 2 km combined with the high-resolution SST and soil datasets. Slight degradation occurred when the satellite GVF dataset was included.

In part II of this study (Pierce et al. 2023), meteorological analyses obtained from the baseline EPA and optimized WRF model configurations are used as input to CMAQ model simulations to assess their impact on ozone forecasts in the Lake Michigan region.

**Acknowledgments**

Funding for this project was provided by the NASA Health and Air Quality (HAQ) program via grant #80NSSC18K1593.

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
