# Peer review of "Meteorological modeling sensitivity to parameterizations and satellite-derived surface datasets during the 2017 Lake Michigan Ozone Study"

_EGUsphere, 2023_

## Referee Comment (RC1)

General comments

This study aims to evaluate different techniques involving ingestion of vegetation, soil temperature and moisture, and SST data from satellite and LDAS output, to improve high-resolution WRF simulations for use with the CMAQ air quality model applied to the LMOS field study. The study also includes two different physics configurations that alter the LSM, PBL, and microphysics. Unfortunately, the model configuration designated "EPA" seems flawed relative to our experiences modeling many fine scale domains. How it may be flawed, however, can only be speculated. I think more should be done before this "EPA" simulation is used. We understand that many users may not fully appreciate some of the details in configuring inputs and some namelist settings (i.e., Obsgrid) that need some scale awareness, but we think that a sensitivity study of different physics configurations should make sure the model has the most appropriate inputs and runs correctly. Instead, the paper seems to suggest that these results are just what you get when running the "EPA" configuration. Also, calling what they have done the "EPA" configuration makes it seem like they were either performed by the US EPA or endorsed. The EPA has used similar configurations at high-resolution with good results as shown in the recently published LISTOS modeling study (Torres-Vazquez et al. 2022). The LISTOS area is just as complex as LMOS with land-sea breeze impacts. The LISTOS modeling showed that the 2-m temperature RMSE for the 1.33 km domain were clearly lower than the 12 km (1.75 K for 12 km and 1.60 K for 1.33 for the whole summer) which is also much lower than the LMOS results shown here. The reasons for the poor results may be related to the data assimilation procedures/inputs used in this study. However, it is not clear from the manuscript what was done in some key areas of data preparation.

It is mentioned that the indirect soil moisture and temperature nudging was used in the PX LSM. Some details are not clear. It seems that they used the 0.25-degree resolution GFS Final re-analyses available at 6-h intervals as background to re-analyze with MADIS observations using OBSGRID. However, these analyses are much too coarse spatially and temporally to give good results for WRF runs at 1.33 km resolution which would explain why the results are not as good as the runs used for LISTOS which used the UnRestricted Mesoscale Analysis (URMA) data set, available in as fine as 1-hourly increments on a 2.5 km grid spacing for 1.33 km WRF runs. Some of these ideas for best practices for running the PX LSM at high resolution are described in a document [http://www2.mmm.ucar.edu/wrf/users/docs/PX-ACM.pdf](http://www2.mmm.ucar.edu/wrf/users/docs/PX-ACM.pdf) referenced in the WRF users' guide [https://github.com/wrf-model/Users_Guide](https://github.com/wrf-model/Users_Guide). However, this alone is unlikely to have resulted in the large errors and biases especially in humidity that are shown in the paper. Therefore, I think there may have been other issues like potentially problems with the observation nudging.

It is stated several places that surface observation nudging was used for the "EPA" runs, but the details of how this was done are not described. On lines 470-472: *The superior performance of the EPA simulation on the 12-km domain is partially an artifact of its use of surface observation nudging because the same observations used in the nudging routine were also used for verification.* If the same observation nudging was used for the 1.33 km runs the results would also show superior performance at the observation locations. Seems something went wrong and likely an issue with the nudging itself. Until the flaws in the "EPA" runs are investigated, discovered, and corrected, I suggest that these simulations be removed from the study. It seems like the experimental design implicitly linked obs nudging to the US EPA configuration. We think this should have been a separate sensitivity.

The comparisons of the various YNT sensitivity simulations show relatively small differences.  There are some interesting analyses of the differences and discussion relating the differences to the variations in the use of soil, vegetation, and SST data sets and a change in the height of analysis FDDA.  However, these discussions repeatedly refer to comparisons to the flawed "EPA" runs.

Specific comments:

Line 20:  Please do not refer to the simulation using PX and ACM2 as "EPA" since the EPA was not involved with these runs.

Line 24:  "physically unrealistic ground heat flux".  This should have been a clue that something was seriously wrong and should be investigated.

Line 93: This is a 7 year old version of WRF.  Why not use a newer version?  There have been significant changes and error corrections in PX and ACM2 since then. Noah has also been updated.

Line 111: misleading.  Better to say local and non-local vertical transport

116:  "available surface observations" is not 100% accurate here. The indirect nudging uses the surface analysis in the wrfsfdda_d0* files. Obsgrid creates the wrfsfdda_d01 file using the GFS-FNL here + MADIS point obs, but it is more accurate to say the indirect soil nudging uses a surface analysis from Obsgrid.

121-123: More explanation is needed especially about the background analysis

144:  If I understood correctly some of these datasets are use daily not just for initialization

Table 1.  Please clarify what is meant by "obs nudging to MADIS".  Is this direct obs nudging or the indirect soil T and SM nudging using obsgrid?

231-233: Again, need to explain surface nudging

270-272:  Basic rule of thumb: don't use coarser analyses to nudge finer models

283-284: This is unreasonable performance.  The simulation should be fixed or removed from comparison.

310-311:  This suggests deficiencies in YNT physics

334: The fact that WS errors were smaller for "EPA" than T2m and q-2m suggests that the data used for soil nudging was very detrimental.

362-365: This shows some serious errors in the "EPA" runs.  Need to figure that out or remove from comparisons.

405: This assumed surface energy balance neglects Cp dTg/dt.  Why not sum the up and down SW and LW components? That would be net surface radiation.

425-434: These results seem to indicate that there is a larger problem than the low resolution of the nudging data.  The Gflux always < 0 and the large wet bias day and night suggests that the nudging data is much too cold.  If a DA scheme uses bad data, the results will be bad.  The surface nudging analyses

should be evaluated the same way as the model output. I expect this will show large errors in the nudging data. This might suggest errors in the OBSGRID processing.

445: This is contrary to what the LISTOS study showed

460: Again, please clarify "surface observation nudging"

462: Noah is not more sophisticated than PX LSM overall. While Noah has more soil layers, PX uses more refined representation of land use related parameters. Also, the indirect soil nudging capability is a big advantage when applied correctly as has been demonstrated many times in many publications. In other ways the two models are roughly equivalent.

464: Again, not just initialization.

470-472: if the good performance of the 12 km "EPA" run is due to obs nudging then why did the 1.33 km run not have similarly good results. Observation nudging using the same obs as used in the statistics should give much, much better results.

474-476: I agree that the relative spatial sparseness of the obs do not accurately capture and constrain small-scale features on the 1.33 km grid but evaluating wth the same observations as used on the obs nudging should result is very close agreement regardless of the model grid scale. This suggest some potential problem with the obs nudging that should be investigated, or removing Obs nudging from the experiment or as an independent sensitivity

485-488: The two-layer soil structure is designed to work with the indirect nudging. This normally works well and gives better results that other LSMs when applied correctly.

---

## Author Comment (AC1)

*We thank Jonathan Pleim for his detailed comments that helped improve the manuscript.*

Reviewer #1:

General comments

This study aims to evaluate different techniques involving ingestion of vegetation, soil temperature and moisture, and SST data from satellite and LDAS output, to improve high-resolution WRF simulations for use with the CMAQ air quality model applied to the LMOS field study. The study also includes two different physics configurations that alter the LSM, PBL, and microphysics. Unfortunately, the model configuration designated "EPA" seems flawed relative to our experiences modeling many fine scale domains. How it may be flawed, however, can only be speculated. I think more should be done before this "EPA" simulation is used. We understand that many users may not fully appreciate some of the details in configuring inputs and some namelist settings (i.e., Obsgrid) that need some scale awareness, but we think that a sensitivity study of different physics configurations should make sure the model has the most appropriate inputs and runs correctly. Instead, the paper seems to suggest that these results are just what you get when running the "EPA" configuration. Also, calling what they have done the "EPA" configuration makes it seem like they were either performed by the US EPA or endorsed. The EPA has used similar configurations at high-resolution with good results as shown in the recently published LISTOS modeling study (Torres-Vazquez et al. 2022). The LISTOS area is just as complex as LMOS with land-sea breeze impacts. The LISTOS modeling showed that the 2-m temperature RMSE for the 1.33 km domain were clearly lower than the 12 km (1.75 K for 12 km and 1.60 K for 1.33 for the whole summer) which is also much lower than the LMOS results shown here. The reasons for the poor results may be related to the data assimilation procedures/inputs used in this study. However, it is not clear from the manuscript what was done in some key areas of data preparation.

It is mentioned that the indirect soil moisture and temperature nudging was used in the PX LSM. Some details are not clear. It seems that they used the 0.25-degree resolution GFS Final re-analyses available at 6-h intervals as background to re-analyze with MADIS observations using OBSGRID. However, these analyses are much too coarse spatially and temporally to give good results for WRF runs at 1.33 km resolution which would explain why the results are not as good as the runs used for LISTOS which used the UnRestricted Mesoscale Analysis (URMA) data set, available in as fine as 1-hourly increments on a 2.5 km grid spacing for 1.33 km WRF runs. Some of these ideas for best practices for running the PX LSM at high resolution are described in a document http://www2.mmm.ucar.edu/wrf/users/docs/PX-ACM.pdf referenced in the WRF users' guide https://github.com/wrf-model/Users_Guide. However, this alone is unlikely to have resulted in the large errors and biases especially in humidity that are shown in the paper. Therefore, I think there may have been other issues like potentially problems with the observation nudging.

It is stated several places that surface observation nudging was used for the "EPA" runs, but the details of how this was done are not described. On lines 470-472: *The superior performance of the EPA simulation on the 12-km domain is partially an artifact of its use of surface observation nudging because the same observations used in the nudging routine were also used for verification.* If the same observation nudging was used for the 1.33 km runs the results would also show superior

performance at the observation locations. Seems something went wrong and likely an issue with the nudging itself. Until the flaws in the "EPA" runs are investigated, discovered, and corrected, I suggest that these simulations be removed from the study. It seems like the experimental design implicitly linked obs nudging to the US EPA configuration. We think this should have been a separate sensitivity.

The comparisons of the various YNT sensitivity simulations show relatively small differences. There are some interesting analyses of the differences and discussion relating the differences to the variations in the use of soil, vegetation, and SST data sets and a change in the height of analysis FDDA. However, these discussions repeatedly refer to comparisons to the flawed "EPA" runs.

*Thank you for your comments. We have changed "EPA" to "AP-XM" throughout the revised manuscript to avoid the appearance that these simulations were performed or endorsed by the EPA. The new naming convention follows that of the "YNT" configuration (e.g., initials for the names of the PBL, LSM, and microphysics schemes). We appreciate your concern about the accuracy of the EPA (AP-XM) simulation. We have confirmed that the namelist options we used for this simulation are correct; however, given the presence of the large moist bias, we cannot rule out that there was a problem with its implementation. Because we are unable to run new simulations at this late stage of the project, we have instead decided to include results from an earlier version of the "EPA" simulation that we had run that did not use observation nudging or soil temperature and soil moisture nudging, which is hereafter referred to as the "AP-XM" simulation. The figure below shows the summary statistics for this simulation along with the old EPA simulation (which is named AP-XM_OBS in the figure) that used observation and soil nudging. Though the temperature and wind speed errors in the AP-XM simulation (without soil and observation nudging) are slightly larger than the AP-XM_OBS simulation (with soil and observation nudging), the water vapor bias and RMSE have been greatly reduced. The overall conclusions from this sensitivity test remain the same as before where the AP-XM configuration provides superior results for 2-m temperature and 10-m wind speed on the 12-km domain but then becomes less accurate than the YNT configuration on the 1.3-km resolution domain, while also having a moist bias and larger water vapor RMSEs on all three domains.*

*Based on this sensitivity test, we have decided to include results from the "AP-XM" simulation without observation and soil nudging in the revised manuscript. This has the notable advantage of allowing us to avoid the situation where the same surface observations used for the evaluation are also used in the observation nudging routine for one of the simulations. This means that the evaluation methods can be applied evenly across all of the model simulations, which strengthens the conclusions of the revised paper. It also eliminates potential biases being introduced though sub-optimal use of the observation and soil nudging schemes. Though the document noted by the reviewer recommends that nudging should be used with the Pleim-Xiu land surface model, the decision not to use it in the AP-XM simulation in this study is supported by the fact that there have been many studies over the past few years that have used the Pleim-Xiu scheme without observation or soil nudging (e.g., Bhautmage et al. 2022, J. Geophys. Res.; Parra 2023, Atmosphere; Parde et al. 2022, Atmos. Res.; Lu et al. 2021, Clim. Dyn.; Jacondino et al. 2021, Energy). Nonetheless, to acknowledge these additional tools, we have added two sentences to the second paragraph in the conclusions section describing results from the AP-XM simulation stating that: "In addition, use of observation nudging and soil moisture and soil temperature*

*nudging as used in Torres-Vazquez et al. (2022) would also help constrain the evolution of this simulation. Though these specialized nudging techniques were not employed in our study due to their added complexity and confounding influence on the model evaluations because the same observations used in the nudging procedure would also be used to assess the accuracy of the simulations, their utility could be assessed in future work."*

|  | **Bias** | | | **% RMSE Change vs. AP_XM_OBS** | | | **% RMSE Change vs. YNT** | | |
|---|---|---|---|---|---|---|---|---|---|
| **a)** 2-m Temperature [K] | | | | **b)** 2-m Temperature [K] | | | **c)** 2-m Temperature [K] | | |
| Simulation | 12 km | 4 km | 1.3 km | 12 km | 4 km | 1.3 km | 12 km | 4 km | 1.3 km |
| AP_XM | -0.66 | -0.85 | -0.14 | 2.27 | 2.36 | 3.03 | | | |
| AP_XM_OBS | -0.12 | -0.40 | 0.16 | 2.03 | 2.23 | 3.00 | | | |
| YNT | 0.16 | 0.47 | 0.55 | 13.08 | 0.45 | -25.18 | 2.30 | 2.24 | 2.25 |
| YNT_SST | 0.17 | 0.48 | 0.56 | 12.44 | -0.13 | -25.58 | -0.57 | -0.58 | -0.53 |
| YNT_SOIL | -0.39 | -0.19 | -0.22 | 11.95 | -4.62 | -30.41 | -1.00 | -5.04 | -6.99 |
| YNT_N2KM | 0.25 | 0.58 | 0.67 | 12.44 | -0.18 | -24.68 | -0.57 | -0.62 | 0.67 |
| YNT_GVF | -0.28 | -0.02 | -0.03 | 12.54 | -1.88 | -27.91 | -0.48 | -2.32 | -3.65 |
| YNT_SSNG | -0.56 | -0.32 | -0.38 | 10.62 | -5.20 | -29.71 | -2.17 | -5.62 | -6.06 |
| YNT_SSN | -0.29 | -0.07 | -0.09 | 9.00 | -7.44 | -31.91 | -3.61 | -7.85 | -8.99 |

|  | **Bias** | | | **% RMSE Change vs. AP_XM_OBS** | | | **% RMSE Change vs. YNT** | | |
|---|---|---|---|---|---|---|---|---|---|
| **d)** 2-m Mixing Ratio [g/kg] | | | | **e)** 2-m Mixing Ratio [g/kg] | | | **f)** 2-m Mixing Ratio [g/kg] | | |
| Simulation | 12 km | 4 km | 1.3 km | 12 km | 4 km | 1.3 km | 12 km | 4 km | 1.3 km |
| AP_XM | 0.38 | 0.64 | 0.60 | 1.67 | 1.80 | 1.70 | | | |
| AP_XM_OBS | 0.91 | 1.28 | 1.35 | 1.85 | 2.03 | 2.07 | | | |
| YNT | 0.19 | 0.00 | -0.20 | -19.96 | -28.69 | -29.85 | 1.48 | 1.44 | 1.45 |
| YNT_SST | 0.20 | 0.00 | -0.20 | -20.66 | -29.19 | -30.48 | -0.88 | -0.69 | -0.90 |
| YNT_SOIL | 0.24 | 0.10 | -0.02 | -20.12 | -29.14 | -31.40 | -0.20 | -0.62 | -2.21 |
| YNT_N2KM | 0.22 | 0.05 | -0.14 | -19.69 | -28.10 | -28.88 | 0.34 | 0.83 | 1.38 |
| YNT_GVF | 0.30 | 0.17 | 0.02 | -20.12 | -28.69 | -30.77 | -0.20 | 0.00 | -1.31 |
| YNT_SSNG | 0.36 | 0.28 | 0.24 | -22.33 | -29.83 | -31.54 | -2.96 | -1.59 | -2.41 |
| YNT_SSN | 0.27 | 0.14 | 0.04 | -21.20 | -29.83 | -32.03 | -1.55 | -1.59 | -3.10 |

|  | **Bias** | | | **% RMSE Change vs. AP_XM_OBS** | | | **% RMSE Change vs. YNT** | | |
|---|---|---|---|---|---|---|---|---|---|
| **g)** 10-m Wind Speed [m/s] | | | | **h)** 10-m Wind Speed [m/s] | | | **i)** 10-m Wind Speed [m/s] | | |
| Simulation | 12 km | 4 km | 1.3 km | 12 km | 4 km | 1.3 km | 12 km | 4 km | 1.3 km |
| AP_XM | -0.02 | -0.22 | -0.23 | 1.51 | 1.50 | 1.62 | | | |
| AP_XM_OBS | 0.05 | -0.17 | -0.14 | 1.52 | 1.51 | 1.63 | | | |
| YNT | 0.45 | 0.34 | 0.36 | 6.26 | 2.19 | -3.32 | 1.61 | 1.54 | 1.57 |
| YNT_SST | 0.46 | 0.34 | 0.36 | 6.52 | 2.52 | -2.40 | 0.25 | 0.32 | 0.95 |
| YNT_SOIL | 0.38 | 0.24 | 0.23 | 5.07 | 1.26 | -4.49 | -1.12 | -0.91 | -1.21 |
| YNT_N2KM | 0.42 | 0.32 | 0.34 | 4.61 | 0.60 | -5.05 | -1.55 | -1.56 | -1.78 |
| YNT_GVF | 0.60 | 0.54 | 0.60 | 10.87 | 7.97 | 4.06 | 4.34 | 5.65 | 7.64 |
| YNT_SSNG | 0.53 | 0.47 | 0.49 | 8.04 | 5.25 | -0.25 | 1.67 | 2.99 | 3.18 |
| YNT_SSN | 0.36 | 0.23 | 0.22 | 3.82 | -0.20 | -6.52 | -2.29 | -2.34 | -3.31 |

Specific comments:

Line 20: Please do not refer to the simulation using PX and ACM2 as "EPA" since the EPA was not involved with these runs.

*We have changed "EPA" to "AP-XM" here and elsewhere in the manuscript.*

Line 24: "physically unrealistic ground heat flux". This should have been a clue that something was seriously wrong and should be investigated.

*Prior to submission of this paper, we verified that we had used the correct settings and run-time options for the "AP-XM" (nee EPA) simulation. We cannot rule out that there could be a bug in the version of the P-X LSM available in version 3.8.1 of the WRF model, but the fact that the ground heat flux has a realistic diurnal pattern and the other surface fluxes are very similar to those of the YNT simulations on the 12-km domain but then becomes unrealistic on the 1.3-km domain points toward limitations in the AP-XM configuration when used at higher resolutions.*

Line 93: This is a 7 year old version of WRF. Why not use a newer version? There have been significant changes and error corrections in PX and ACM2 since then. Noah has also been updated.

*This project began in late 2018. The version of the WRF model that we used during this study was just over a year old at the start of the project. To maintain consistency, we decided to keep using that version of the model throughout the project. We were planning to submit this paper during 2020 but then decided to wait until we could submit it along with our evaluation of the CMAQ model simulations (part II of this study). That decision, along with delays due to covid, meant that we did not submit this paper until earlier this year.*

Line 111: misleading. Better to say local and non-local vertical transport

*This sentence has been revised as requested.*

116: "available surface observations" is not 100% accurate here. The indirect nudging uses the surface analysis in the wrfsfdda_d0* files. Obsgrid creates the wrfsfdda_d01 file using the GFS-FNL here + MADIS point obs, but it is more accurate to say the indirect soil nudging uses a surface analysis from Obsgrid.

*This sentence has been removed because we are no longer using observation nudging.*

121-123: More explanation is needed especially about the background analysis

*This sentence has been removed from the revised manuscript because we are no longer using observation nudging in the AP-XM simulation.*

144: If I understood correctly some of these datasets are use daily not just for initialization

*Thank you for the comment. We have decided to refer to these as "surface datasets" rather than "surface initialization datasets" because, as you point out, they are used not only to initialize the model but also to update the surface fields each day. Revisions have been made throughout the manuscript to account for this change.*

Table 1. Please clarify what is meant by "obs nudging to MADIS". Is this direct obs nudging or the indirect soil T and SM nudging using obsgrid?

*This statement has been removed because we are no longer using observation nudging in the AP-XM simulation.*

231-233: Again, need to explain surface nudging

*This sentence has been removed because we are no longer using observation nudging in the AP-XM simulation.*

270-272: Basic rule of thumb: don't use coarser analyses to nudge finer models

*This sentence has been deleted from the revised manuscript because we are no longer using observation nudging with the AP-XM simulation.*

283-284: This is unreasonable performance. The simulation should be fixed or removed from comparison.

*We have replaced the EPA simulation with the AP-XM simulation that does not use observation or soil nudging. For 2-m temperature, the AP-XM simulation provides superior performance on the 12-km domain and comparable performance on the 4-km domain, but was less accurate on the 1.3-km domain when compared to the YNT simulations.*

310-311: This suggests deficiencies in YNT physics

*We have added this sentence to the end of this paragraph: "This behavior could also be due to deficiencies in the YNT configuration over complex urban-lake transition zones."*

334: The fact that WS errors were smaller for "EPA" than T2m and q-2m suggests that the data used for soil nudging was very detrimental.

*We state in the second paragraph of Section 3.1 in the revised manuscript that the AP-XM run contains the smallest wind speed bias on all three domains and lower RMSE on the outermost two domains when compared to the YNT simulations.*

362-365: This shows some serious errors in the "EPA" runs. Need to figure that out or remove from comparisons.

*We have replaced the EPA simulation with the AP-XM simulation that does not use observation nudging or soil nudging. The water vapor errors are much smaller now, though the temperature and wind speed errors are similar to what was obtained during the EPA simulation.*

405: This assumed surface energy balance neglects Cp dTg/dt. Why not sum the up and down SW and LW components? That would be net surface radiation.

*Thank you for the suggestion. We have revised Fig. 7b so that the net radiation is now computed using the sum of the upward and downward shortwave and longwave radiation components.*

425-434: These results seem to indicate that there is a larger problem than the low resolution of the nudging data. The Gflux always < 0 and the large wet bias day and night suggests that the nudging data is much too cold. If a DA scheme uses bad data, the results will be bad. The surface nudging analyses should be evaluated the same way as the model output. I expect this will show large errors in the nudging data. This might suggest errors in the OBSGRID processing.

*The revised manuscript includes results from the AP-XM simulation that did not use observation or soil nudging. The overall pattern remains the same to what was found in the original "EPA" simulation that used observation and soil nudging, namely that the diurnal cycle of the ground heat flux was similar to the YNT simulations on the 12-km domain but was consistently negative on the 1.3 km domain. This points toward this behavior not being due to the coarse resolution of the nudging data.*

445: This is contrary to what the LISTOS study showed

*This sentence referred to the behavior of the surface flux terms during the AP-XM simulation. Torres-Vazquez et al. (2022) did not evaluate the accuracy of these terms in their paper.*

460: Again, please clarify "surface observation nudging"

*Surface observation nudging is no longer used during the AP-XM simulation.*

462: Noah is not more sophisticated than PX LSM overall. While Noah has more soil layers, PX uses more refined representation of land use related parameters. Also, the indirect soil nudging capability is a big advantage when applied correctly as has been demonstrated many times in many publications. In other ways the two models are roughly equivalent.

*This sentence has been removed from the revised manuscript.*

464: Again, not just initialization.

*As mentioned earlier, we now refer to these datasets as "surface datasets" rather than "surface initialization datasets" because, as you point out, they are used not only to initialize the model but also to update the surface fields each day.*

470-472: if the good performance of the 12 km "EPA" run is due to obs nudging then why did the 1.33 km run not have similarly good results. Observation nudging using the same obs as used in the statistics should give much, much better results.

*The revised manuscript includes results from the AP-XM run that does not employ observation nudging. With this new run, there is still a moist bias on all three domains, but it is much smaller than occurred when using observation nudging.*

474-476: I agree that the relative spatial sparseness of the obs do not accurately capture and constrain small-scale features on the 1.33 km grid but evaluating with the same observations as used on the obs nudging should result is very close agreement regardless of the model grid scale. This suggest some potential problem with the obs nudging that should be investigated, or removing Obs nudging from the experiment or as an independent sensitivity

***The revised manuscript includes results from the AP-XM run that does not employ observation nudging.***

485-488: The two-layer soil structure is designed to work with the indirect nudging. This normally works well and gives better results that other LSMs when applied correctly.

***We have added two sentences to the end of this paragraph noting that: "In addition, use of observation nudging and soil moisture and soil temperature nudging as used in Torres-Vazquez et al. (2022) would also help constrain the evolution of this simulation. Though these specialized nudging techniques were not employed in our study due to their added complexity and confounding influence on the model evaluations because the same observations used in the nudging procedure would also be used to assess the accuracy of the simulations, their utility could be assessed in future work."***

---

## Author Comment (AC2)

*We thank the anonymous reviewer for their detailed comments that helped improve the manuscript.*

Reviewer #2:

In this study, the authors performed eight WRF model simulations (at 12, 4, and 1.3 km horizontal resolutions) to assess the impact of different parameterization schemes, land/lake surface initialization approaches, and analysis nudging methods on the simulated surface energy fluxes and near-surface atmospheric conditions over the Lake Michigan region during the 1-month LMOS field campaign period in 2017. Model evaluation presented in this work helped the same group to select meteorological inputs of the CMAQ simulations described in a companion paper by Pierce et al., which is also currently under review for the same journal.

My major comments include:

**Novelty:** Comparing Pleim-Xiu with indirect soil nudging and Noah is not a new idea-more than 7 years ago, a TCEQ funded project on such a topic was conducted. Many modeling communities including at NASA and NOAA plan to (or are already actively working on) migrating from Noah to Noah-MP land surface model due to known limitations in Noah. Initialization WRF using output from LIS or similar frameworks to benefit weather and air quality studies is not new neither, which has been recognized by the authors themselves. Running at very high-resolution over regions with complex surface types (e.g., land vs water) is also broadly appreciated by modeling communities. Although this is a companion paper of Pierce et al., it should also be able to stand alone with its own highlights. Thus the author are encouraged to clearly underscore the novel aspects of this study, and this may lead to adding some additional modeling experiments and more rigorous evaluation. And perhaps the expected paper would fit better into GMD. Otherwise, shortening the paper and merging the key information into Pierce et al. is suggested.

*The novel contribution of this paper is its detailed assessment of a multi-resolution (12 km to 1.3 km) and multi-constraint modeling framework for the Lake Michigan area. As discussed in the introduction, given the important role that boundary layer meteorology and the land-lake breeze circulation have on ozone production and transport in this region, it is critical to explore the ability of different parameterization schemes and surface datasets to improve the accuracy of near-surface meteorological and air quality simulations. Regarding the Pierce et al. (2023) companion paper, we have added some text to better tie these two papers together. This includes describing the importance of surface meteorology on ozone production in the last paragraph of the introduction and discussing how differences in meteorology impact biogenic emissions in the chemistry simulations. We have also added a sentence to the last paragraph of this paper to serve as a segue to the Pierce et al. (2023) paper. It is not feasible to combine these two papers into a single paper given the amount of material being presented; however, we are willing to transfer this paper to GMD, if necessary, as long as it does not require another set of reviews.*

**Methods and presentation:** While a lot of information is given, clarifications on the methods are still necessary. Specifically:

1. The authors stated at L222-223 that "*Direct insertion into the WRF model was possible because of the similarly configured Noah LSM used in both the LIS and WRF simulations*". Please provide the version of Noah in LIS that was used in this study as well as evidence showing it's similar to Noah embedded in WRF3.8.1. Also, this statement cannot be agreed if the static inputs (land use/land cover, soil type, terrain, etc) of the land model are consistent in LIS/Noah and WRF3.8.1. LIS and WRF3.8.1 static inputs are generated from LDT and WPS tools, respectively. Please clarify what exactly has been done. This study area has complex surface type (not only land vs water, but also for land, urban vs non-urban categories), so some discussions on how these surface characteristics are represented by the model would be very informative.

*Our use of "direct insertion" in lines 222-223 when describing how we used the SPoRT LIS soil moisture and soil temperature data was imprecise. It was simply meant to convey that the use of the same vertical layers in the Noah LSM used in both SPoRT LIS and the WRF model made it easier to use the LIS output in our WRF model sensitivity experiments since there would have been no reason to interpolate between vertical layers. We have deleted this sentence from the revised paper because it was unnecessary. Version 3.6 of the Noah LSM was used in the SPoRT LIS, which is very similar to that used in version 3.8.1 of the WRF model. All of our simulations with the Noah LSM used the default settings for the vegetation and soil properties, with the exception of the YNT_GVF and YNT_SSNG simulations where the climatological vegetation fraction was replaced by the high-resolution daily VIIRS green vegetation fraction data. To address your request for more information, we have expanded the first paragraph of the methods section describing the WRF model configurations to provide additional detail about how the simulations were performed and to mention that the WRF Preprocessing System (WPS) was used to prepare the surface datasets. In particular, we have added these sentences: "Except for the two baseline simulations described below, all of the simulations were performed in daily increments using the standard WRF model restart files to allow for daily updates of high-resolution surface datasets using the WPS. A sentence has also been added to Section 2.1 stating that: "The 40-category National Land Cover Dataset (NLCD) 2011 land use dataset (Jin et al. 2013) was used to determine the vegetation type and soil properties for each model grid point."*

2. The evaluation of the model runs are not rigorous and not well connected with Pierce et al. The uncertainty of VIIRS GVF is not introduced in the paper - this product sounds to have short latency but for retrospective analysis like it'd be important to tell its quality on a daily time scale for this region relevant to air pollution events studied. Similar comment on GLSEA. The SPoRT LIS product is not discussed clearly (it is not very clear whether land data assimilation is enabled in the LIS system and if so, some data assimilation diagnostics could be shown) - my understanding is that SPoRT hosts documentations and visualizations of these routinely generated products elsewhere which may be cited in the paper. In terms of WRF model evaluation, some statistics and maps are presented but only for a limited number of variables, and as the authors noted at L232-233, "*these surface observations were also used to perform surface nudging during the EPA simulation, which will impact the results presented in Section 3 because surface nudging was not used during any of the YNT simulations*". As the model outputs served as meteorological input of CMAQ, a list of variables central to pollutants to be studied should be selected with

justifications, followed by model performance of them. The performance could be discussed in connection with the air pollution events and time series presented in Pierce et al., and additional evaluation metrics such as correlations between modeled and observed time series may be added. Furthermore, are there really no in-situ flux measurements/PBL info across the entire three WRF domains as stated at L402?

*To address your comment about the VIIRS and GLSEA datasets, we have added sentences to Sections 2.2.1 and 2.2.2 stating that: "Only satellite observations are used to produce the daily lake surface temperature analyses, which Schwab et al. (1992) showed have small bias and root mean square error (1-1.5 °C) when compared to buoys." and "Ding and Zhu (2018) have shown that the VIIRS GVF product has smaller errors and bias than other satellite derived GVF datasets because of reduced atmospheric influences, improved observing capabilities in high biomass regions, better representation of vegetation canopies, and reduced bidirectional reflection distribution function effects." Regarding the NASA SPoRT LIS run, we now state in Section 2.2.3 of the revised manuscript that no observations were assimilated during the LIS runs. The model variables that we chose to evaluate in this paper (2-m temperature, 2-m water vapor mixing ratio, 10-m wind speed, and PBL height) are all important for air quality modeling applications. They are typically used in model verification studies given their relevance and availability over large spatial domains. As for flux tower measurements, there are a few stations across the domain; however, we have chosen not to include them in this analysis because of their sparse distribution and difficulties handling representativeness issues due to differences in spatial scale. Surface fluxes can vary greatly over short distances, which makes it difficult to use them for model verification. Finally, as requested, we computed the correlations between the simulated and observed meteorological variables using hourly data over the 7-week study period. The correlations are shown in the right column in the figure below. It is evident that the correlations are very similar among all of the simulations for 2-m temperature, 2-m mixing ratio, and 10-m wind speed on the 12-km and 4-km domains. On the 1.3-km domain, however, the correlations are less for the AP-XM simulation whereas they remain similarly high among the various YNT simulations. These results are consistent with what was shown in the RMSE percentage statistics shown in the original manuscript and therefore we decided not to include the correlations in the revised manuscript.*

| Simulation | Bias a) 2-m Temperature [K] 12 km | 4 km | 1.3 km | % RMSE Change vs. AP_XM b) 2-m Temperature [K] 12 km | 4 km | 1.3 km | % RMSE Change vs. YNT c) 2-m Temperature [K] 12 km | 4 km | 1.3 km | Correlation d) 2-m Temperature [K] 12 km | 4 km | 1.3 km |
|---|---|---|---|---|---|---|---|---|---|---|---|---|
| AP_XM | -0.66 | -0.83 | -0.14 | 2.27 | 2.36 | 3.03 | | | | 0.93 | 0.92 | 0.85 |
| YNT | 0.16 | 0.47 | 0.55 | 1.37 | -5.12 | -25.83 | 2.30 | 2.24 | 2.25 | 0.93 | 0.93 | 0.93 |
| YNT_SST | 0.17 | 0.48 | 0.56 | 0.79 | -5.67 | -26.22 | -0.57 | -0.58 | -0.53 | 0.93 | 0.93 | 0.93 |
| YNT_SOIL | -0.39 | -0.19 | -0.22 | 0.35 | -9.91 | -31.01 | -1.00 | -5.04 | -6.99 | 0.93 | 0.93 | 0.93 |
| YNT_N2KM | 0.25 | 0.58 | 0.67 | 0.79 | -5.72 | -25.33 | -0.57 | -0.62 | 0.67 | 0.93 | 0.93 | 0.93 |
| YNT_GVF | -0.28 | -0.02 | -0.03 | 0.88 | -7.32 | -28.53 | -0.48 | -2.32 | -3.65 | 0.93 | 0.93 | 0.93 |
| YNT_SSNG | -0.56 | -0.32 | -0.38 | -0.84 | -10.46 | -30.32 | -2.17 | -5.62 | -6.06 | 0.93 | 0.93 | 0.93 |
| YNT_SSN | -0.29 | -0.07 | -0.09 | -2.29 | -12.57 | -32.50 | -3.61 | -7.85 | -8.99 | 0.93 | 0.94 | 0.94 |

| Simulation | Bias e) 2-m Mixing Ratio [g/kg] 12 km | 4 km | 1.3 km | % RMSE Change vs. AP_XM f) 2-m Mixing Ratio [g/kg] 12 km | 4 km | 1.3 km | % RMSE Change vs. YNT g) 2-m Mixing Ratio [g/kg] 12 km | 4 km | 1.3 km | Correlation h) 2-m Mixing Ratio [g/kg] 12 km | 4 km | 1.3 km |
|---|---|---|---|---|---|---|---|---|---|---|---|---|
| AP_XM | 0.38 | 0.64 | 0.60 | 1.67 | 1.80 | 1.70 | | | | 0.90 | 0.90 | 0.89 |
| YNT | 0.19 | 0.00 | -0.20 | -10.98 | -19.87 | -14.86 | 1.48 | 1.44 | 1.45 | 0.90 | 0.90 | 0.90 |
| YNT_SST | 0.20 | 0.00 | -0.20 | -11.76 | -20.42 | -15.62 | -0.88 | -0.69 | -0.90 | 0.90 | 0.90 | 0.90 |
| YNT_SOIL | 0.24 | 0.10 | -0.02 | -11.16 | -20.37 | -16.74 | -0.20 | -0.62 | -2.21 | 0.90 | 0.90 | 0.90 |
| YNT_N2KM | 0.22 | 0.05 | -0.14 | -10.68 | -19.20 | -13.68 | 0.34 | 0.83 | 1.38 | 0.90 | 0.90 | 0.90 |
| YNT_GVF | 0.30 | 0.17 | 0.02 | -11.16 | -19.87 | -15.97 | -0.20 | 0.00 | -1.31 | 0.91 | 0.91 | 0.90 |
| YNT_SSNG | 0.36 | 0.28 | 0.24 | -13.62 | -21.14 | -16.91 | -2.96 | -1.59 | -2.41 | 0.91 | 0.91 | 0.91 |
| YNT_SSN | 0.27 | 0.14 | 0.04 | -12.36 | -21.14 | -17.50 | -1.55 | -1.59 | -3.10 | 0.90 | 0.90 | 0.90 |

| Simulation | Bias i) 10-m Wind Speed [m/s] 12 km | 4 km | 1.3 km | % RMSE Change vs. AP_XM j) 10-m Wind Speed [m/s] 12 km | 4 km | 1.3 km | % RMSE Change vs. YNT k) 10-m Wind Speed [m/s] 12 km | 4 km | 1.3 km | Correlation l) 10-m Wind Speed [m/s] 12 km | 4 km | 1.3 km |
|---|---|---|---|---|---|---|---|---|---|---|---|---|
| AP_XM | -0.02 | -0.22 | -0.23 | 1.51 | 1.50 | 1.62 | | | | 0.71 | 0.71 | 0.64 |
| YNT | 0.45 | 0.34 | 0.36 | 7.10 | 2.46 | -3.26 | 1.61 | 1.54 | 1.57 | 0.69 | 0.70 | 0.69 |
| YNT_SST | 0.46 | 0.34 | 0.36 | 7.37 | 2.80 | -2.34 | 0.25 | 0.32 | 0.95 | 0.69 | 0.70 | 0.69 |
| YNT_SOIL | 0.38 | 0.24 | 0.23 | 5.91 | 1.53 | -4.43 | -1.12 | -0.91 | -1.21 | 0.69 | 0.70 | 0.69 |
| YNT_N2KM | 0.42 | 0.32 | 0.34 | 5.44 | 0.87 | -4.99 | -1.55 | -1.56 | -1.78 | 0.70 | 0.71 | 0.70 |
| YNT_GVF | 0.60 | 0.54 | 0.60 | 11.75 | 8.26 | 4.13 | 4.34 | 5.65 | 7.64 | 0.69 | 0.70 | 0.69 |
| YNT_SSNG | 0.53 | 0.47 | 0.49 | 8.90 | 5.53 | -0.18 | 1.67 | 2.99 | 3.18 | 0.70 | 0.71 | 0.70 |
| YNT_SSN | 0.36 | 0.23 | 0.22 | 4.65 | 0.07 | -6.47 | -2.29 | -2.34 | -3.31 | 0.70 | 0.71 | 0.70 |

3. More justifications on the design of model configurations should be added: although the focus of the study is on land initialization/model and nudging, the selections of all the other physics, ICs/BCs, and the distributions of vertical layers (40 layers, is this fine enough?) to study this area should be justified, particularly, are the Noah-related setups based on literature or recommendations from any of their partnering local agency? Also, some extended discussions on ACM2 PBL scheme vs YSU scheme and how they affect the different model runs and conclusions would be very helpful.

*Thank you for your comments. We mention in the revised paper that seven of the 40 vertical layers are located below 2 km. Though the number of vertical layers is relatively small, this choice was made to reduce computational expense due to the large number of simulations, domain size, and simulation length used during this project (including CMAQ simulations described in the Pierce et al. companion paper). The model configurations used during this study, including the number of vertical layers, were determined based on feedback from our partners. We also noted in the original manuscript that: "This particular set of schemes was chosen based on our previous studies showing that they performed well during the warm season across the United States (e.g., Harkey and Holloway 2013; Cintineo et al. 2014; Greenwald et al. 2016; Griffin et al. 2021; Henderson et al. 2021). Because there are dozens of parameterization schemes to choose from in the WRF model, we do not aim to find necessarily the best physics suite but instead to assess the potential of using other schemes*

*to improve upon the performance of the baseline AP-XM configuration."* Finally, because multiple parameterization schemes were changed when switching from the AP-XM to YNT baseline simulations, it is impossible to determine how the ACM2 and YSU PBL schemes by themselves affected the simulations. However, to address your comment, we have added a sentence to the final paragraph in Section 3.2.5 (surface energy budget constraints) in the revised manuscript stating: *"Though it is not the focus of this research, differences in PBL height between the AP-XM and YNT simulations could be due to differences in vertical mixing strength and entrainment flux in the AMC2 and YSU PBL schemes (Hu et al. 2010)."*

Minor comments:

*Pleim-Xu* should be Pleim-Xiu throughout the paper

**Thank you for noticing this spelling mistake. We have revised the spelling throughout the paper.**

Table 1: *IC/LC* should be IC/BC

**We have revised this as suggested.**

Using SST as the short form of lake surface temperature is a little confusing

**We agree that this can be confusing, however, this is the naming convention that is used for this dataset (https://coastwatch.glerl.noaa.gov/erddap/griddap/GLSEA_GCS.html).**

The authors defined soil moisture/soil temperature as SOIL but still use soil moisture and (/) soil temperature in multiple places

**We defined this as "SOIL" in the context of the shortened simulation name (YNT_SOIL) and in the abstract. We prefer to explicitly refer to the soil moisture and soil temperature variables in the rest of the manuscript.**

I think using "evaluation" instead of analysis in many places of this paper would be less confusing

**Thank you for the suggestion. We have changed "analysis" to "evaluation" in various locations (such as the titles of the subsections) to avoid confusion with "analysis nudging" and the input data analyses.**

Abstract is very descriptive and specific to this modeling experiment, rather than delivering messages that could impact a broader audience.

**A sentence was added to the end of the abstract stating that: "These results demonstrate the value of using high-resolution satellite-derived surface datasets in model simulations."**

L208: spell out NLDAS-2

**This acronym was already spelled out in the original text.**

Units of Figure 2 differences plot are missing. Text in Figures 7-9 are small.

***Thank you for noticing that the units were missing from the figure caption for Fig. 2. They have been added to the figure in the revised manuscript. We have also increased the font size for this figure, as well as for Figs. 7-9.***